behaviour

extractive foraging, otters, play, rock juggling

**Author for correspondence:**
Mari-Lisa Allison
e-mail: ma563@exeter.ac.uk

# The drivers and functions of rock juggling in otters

Mari-Lisa Allison, Rebecca Reed, Emile Michels and Neeltje J. Boogert

Centre for Ecology and Conservation, University of Exeter, Penryn Campus, Penryn, Cornwall TR10 9FE, UK

  M-LA, 0000-0001-7185-2719; NJB, 0000-0002-1337-4365

Object play refers to the seemingly non-functional manipulation of inanimate items when in a relaxed state. In juveniles, object play may help develop skills to aid survival. However, why adults show object play remains poorly understood. We studied potential drivers and functions of the well-known object play behaviour of rock juggling in Asian small-clawed (*Aonyx cinereus*) and smooth-coated (*Lutrogale perspicillata*) otters. These are closely related species, but Asian small-clawed otters perform extractive foraging movements to exploit crabs and shellfish while smooth-coated otters forage on fish. We thus predicted that frequent rock jugglers might be better at solving extractive foraging puzzles in the first species, but not the latter. We also assessed whether species, age, sex and hunger correlated with rock juggling frequency. We found that juvenile and senior otters juggled more than adults. However, rock juggling frequency did not differ between species or sexes. Otters juggled more when 'hungry', but frequent jugglers did not solve food puzzles faster. Our results suggest that rock juggling may be a misdirected behaviour when hungry and may facilitate juveniles' motor development, but it appears unrelated to foraging skills. We suggest future studies to reveal the ontogeny, evolution and welfare implications of this object play behaviour.

## 1. Introduction

Until recently, it has been difficult to formally define play behaviour [1]. For decades, it was identified simply through interpreting animals' behaviours as indicative of having 'fun'. This led to play being viewed through a highly anthropocentric lens [2]. Recently, five criteria have been identified to establish an objective and widely accepted definition [3]: 'repeated, seemingly non-functional behaviour differing from more adaptive versions structurally, contextually or developmentally, and initiated when the animal is in a relaxed, unstimulating or low-stress setting' [4, p. 91].

Play can be energetically expensive and may present an increased risk of injury. For playfulness to be selected for, the behaviour must thus provide a significant benefit that outweighs any potential cost [5]. However, the question of the adaptive function of play has long been a source of confusion and debate. Progress in addressing why animals play has been delayed owing to a combination of factors; in addition to the long-standing lack of consensus concerning a definition, the rare and sporadic nature of occurrences of play behaviour in wild animals make it difficult to collect large amounts of data. Even in domestic species that frequently play, studying the behaviour remains problematic. Deprivation studies are typically used to investigate the importance and function of a behaviour. However, depriving an animal of playing often leads to both practical and ethical concerns [6]. Play behaviour is also incredibly varied [1], so it is unlikely that a single type of benefit can account for all play behaviour seen across all species and contexts. As a result, as many as 30 different hypotheses have been proposed in efforts to explain play [7]. Among this diversity, three primary forms of play have been identified: locomotor, social, and object play.

Locomotor play is a solitary behaviour involving intense or sustained body movements [8]. The 'motor-training hypothesis' [9] suggests that this form of play aids neuromuscular development [10]. The benefit of locomotor play in juveniles is thought to appear later in life. For example, young gazelles seem to play by running, pronking and stotting, movements similar to those used by adults when evading predators [11]. On a proximate level, locomotor play is likely to benefit motor development by influencing synapse formation and the differentiation of skeletal muscle fibre types [10].

Social play encompasses all play behaviour directed towards other animate beings, typically conspecifics [12]. This form of play can aid the formation of social bonds and influence dispersal patterns in mature individuals [13]. However, as social play often features locomotor elements, it can share and even enhance the benefits offered by locomotor play. For example, a study on juvenile Belding's ground squirrels (*Urocitellus beldingi*) suggests that higher rates of social play (i.e. play fighting) can greatly improve motor skills [14]. This is owing to these encounters creating novel situations that require one individual to react to the actions of another, which may fine-tune motor movements and increase behavioural flexibility [14].

Object play has been described as 'divertive interactions with inanimate and inedible objects (…) including exploratory manipulation' [15,16, p. 45]. Object play ranges in complexity with some species demonstrating intricate object manipulation or even combinatory action patterns [17], which has led some to suggest that object play may be associated with high intelligence [18]. As a result, object play studies on non-domestic animals tend to be focused on species that are perceived to have enhanced cognitive abilities such as apes, cetaceans and cephalopods [19–21].

The current study aims to increase our understanding of object play by focusing on an understudied taxon that could serve as an ideal system for understanding drivers and functions of object play: otters. Although elusive in the wild, otters are noted to be very playful and inquisitive animals based on observations in captivity [22]. Previous studies of play behaviour in otters have mostly focused on locomotor or social play [23–25]. However, there are several reports of object play behaviour in various otter species [26–28]. The most commonly observed object play behaviour in otters is 'rock juggling', which we define as fast, erratic movements that pass an object between the forepaws and sometimes the mouth. The behaviour is most obvious when performed in a reclined position but may be performed in other stances such as standing upright (see the electronic supplementary material, videos S1 and S2). As there is such little research on this behaviour, there are no formal hypotheses as to the drivers or functions of rock juggling behaviour, other than it being observed more often in hungry than in satiated otters in captivity [29]. To address this gap in our understanding, we studied two otter species commonly found in zoos and wildlife centres and reported to show rock juggling behaviour: Asian small-clawed otters (*Aonyx cinereus*) [30] and smooth-coated otters (*Lutrogale perspicillata*) [28].

Firstly, we investigated whether Asian small-clawed and smooth-coated otters differ in rock juggling frequency in captivity. Although these species are phylogenetically closely related [31], they show numerous morphological and ecological differences. Asian small-clawed otters are the smallest species of otter [32], measuring 0.7–0.93 m in length and weighing 2.7–5.4 kg [33]. They have minimal webbing and claws that do not extend further than the digit, allowing them to manipulate objects with great dexterity [32,34]. Their diets primarily consist of crustaceans and molluscs, foods that require extractive foraging behaviours [35]. Smooth-coated otters are much larger than Asian short-clawed otters, measuring 1.07–1.3 m and weighing 7.0–11.4 kg [33]. They have a greater degree of interdigital webbing [32] and are primarily piscivorous [35], and thus rely less on dextrous movements as they do not need to extract their food. As rock juggling features fine motor movements, we predicted to observe it more often in the Asian small-clawed otters, which naturally perform more dextrous extractive foraging behaviour, when compared with smooth-coated otters.

Secondly, we investigated whether otter age is correlated with rock juggling frequency. Play is prevalent in juvenile animals [6] and is thought to allow them to practice behaviours crucial for adulthood in a relatively safe environment [36]. Furthermore, this peak in object play behaviour in juveniles correlates with a sensitive period of motor neural development characterized by increased cerebral activity and synapse formation [5,37]. Even in species where play persists into adulthood, such as Japanese macaques, there is evidence to suggest that juveniles will engage in play more frequently than adults [37]. We therefore predicted that juvenile otters would rock juggle more frequently than adults.

Thirdly, we addressed whether the sexes differ in rock juggling frequency. Juvenile males in various species display increased levels of social locomotor play behaviour, e.g. play fighting, which is thought to relate to the increased intrasexual competition they will encounter in adulthood [38–40]. With regard to object play, juvenile female chimpanzees (*Pan troglodytes*) perform more stick-holding behaviours than juvenile males, a behaviour thought to be associated with parental care activities, which in turn is more pronounced in females than males [41]. In otters, rock juggling is thought to relate to foraging behaviour. However, foraging is not a sex-specific behaviour; all individuals must provision themselves [35]. As such, we predicted that there would be no significant difference in rock juggling frequency between the sexes.

Fourthly, we investigated how otter hunger levels affected rock juggling frequency. Pellis [29] found that captive Asian small-clawed otters played with objects significantly more before being fed than when they were satiated. Object play behaviour might, therefore, be a form of misdirected foraging. However, these findings were based on observations of a single otter group ($n = 12$), in which all individuals were close relatives. Here, we expand this research by studying multiple groups of otters of two different species and predicted to replicate the original finding that otters will rock juggle more when hungry than when satiated.

Finally, we investigated whether rock juggling is correlated with improved food extraction skills. When studying play behaviour, parallels are often drawn between 'functional' behaviours and the 'incomplete' version observed as play [42]. For example, a cat may chase a ball of yarn using similar motor patterns as would be displayed when hunting mice [5,43]. The motor actions of otters when rock juggling display similarities to those observed when handling prey requiring extraction, such as molluscs. Rock juggling might therefore be relevant to foraging behaviour. Rock juggling is unusual in that it persists into adulthood, suggesting that benefits of the behaviour beyond the developmental context must be considered. Rock juggling may not only be a mechanism by which juvenile otters learn vital motor skills, but it may also allow adult otters to maintain and improve those motor skills. We therefore predicted that otters that performed rock juggling with greater frequency would be faster at extracting food from novel extractive food puzzles.

# 2. Material and methods

## 2.1. Study sites and test subjects

We collected data at three sites in the UK: New Forest Wildlife Park, Newquay Zoo and Tamar Otter and Wildlife Centre. At New Forest Wildlife Park, we studied four groups of Asian small-clawed otters ('ASC'; $n = 4, 4, 2, 1$) and two groups of smooth-coated otters ('SCO': $n = 4, 2$). Newquay Zoo held one group of Asian small-clawed otters ($n = 12$). Tamar Otter and Wildlife Centre had three groups of Asian small-clawed otters ($n = 15, 3, 3$). Across the three sites, we studied a total of 23 males (ASC: $n = 21$; SCO: $n = 2$) and 27 females (ASC: $n = 23$; SCO: $n = 4$). Ages ranged from 6 months to 19 years for Asian small-clawed otters and 3 months to 5 years for smooth-coated otters (for group compositions, see electronic supplementary material, table S1).

To address the research questions on individual and species differences in rock juggling frequency, we first collected observational data on each individual to determine when and how often rock juggling was performed. To address whether rock juggling facilitates extractive foraging behaviour, we then presented each otter group with novel extractive food puzzles to solve.

## 2.2. Rock juggling observations

Prior to data collection, we carried out preliminary observations to compile an ethogram (electronic supplementary material, table S2). Data collection dates and times differed between study sites owing to the location and availability of the establishment and feeding times. At New Forest Wildlife park, we collected data between 5 August and 4 September 2018, between 10.00–14.00 h. At Newquay Zoo, we collected data between 17 October 2018 and 14 February 2019, between 10.30–12.30 h and 14.30–

16.30 h. At Tamar Otter and Wildlife Centre, we collected data between 10 December 2018 and 15 January 2019, between 09.00–16.30 h. We identified otters using differences in body size, shape, fur colour and nose patterns. We conducted observational scans every 3 min over a 1 h period. This was repeated to give a total of 12 h of observation per otter. To assess the influence of hunger levels, we constructed a schedule to observe each otter prior, during and post the establishment's regular feeding times (electronic supplementary material, table S3). Otters were deemed to be satiated for 2 h post feed, as food is reported to take 2 h to pass from mouth to spraint in otters [44].

## 2.3. Den camera trap

To ensure we did not bias our rock juggling observations by only watching otter behaviour outdoors, a motion-triggered camera trap (Bushnell NatureView HD Essential) was installed in the indoor enclosure at Newquay Zoo between 25 and 28 July 2018. Each motion-triggered video recording (resolution 640 × 480p) was 15 s in length and contained a date and time stamp (see the electronic supplementary material, video S2). The 'hunger level' of the otters could thus be inferred by time since last feed. Owing to otters usually being only in partial view, and not in the context of the social group, we could not identify (ID) the otters. Body size was used to differentiate adults from pups; otters less than or equal to 1 year old were considered pups. We scored the number of adult otters and pups present, and whether individuals were rock juggling for each video clip.

## 2.4. Novel extractive food puzzles

We designed three novel extractive food puzzles to quantify individual variation in extractive foraging performance. These puzzle types were designed to be novel to the otters, to control for prior experiences that could influence results. The puzzles were as follows: (i) 50 ml white opaque screw-top medicine bottles (radius = ~2 cm, height = ~7 cm including lid); (ii) tennis balls with an 8 cm diameter cross cut in, with a 1.5 × 1.5 cm square hole in the middle of the cross; and (iii) pairs of bright green Duplo® bricks (each brick measuring 3.1 cm × 6.4 cm × 2.4 cm) stacked on top of each other (figure 1).

We filled each puzzle with 10 g of 5% fat lean minced meat. We placed mince inside the bottles and loosely secured the lid to test general dexterity and digit strength. The same experimenter closed all bottles to ensure consistency. We placed mince inside the hole of the tennis balls to replicate foraging in nooks and crannies. We stuffed mince between the upper and lower Duplo® brick to mimic the extraction of mussels or clams. We also placed monkey nut shells between the bricks to prevent the bricks from being closed too tightly by the otters. This produced a consistent gap of approximately 1 mm between the bricks.

The puzzle types were presented to each group in a different order, i.e. each puzzle type was presented first, second and last at least once across the different otter groups to prevent order effects biasing task solving performance owing to increasing experience across tasks. We provided twice as many individual puzzles as there were otters in each study group (e.g. groups of four otters were presented with eight puzzles) to prevent individuals from monopolising the puzzles and increase the opportunity for all otters to solve the puzzles. Puzzles were dropped in at three separate points at each otter enclosure to further facilitate all individuals accessing the puzzles and to aid in individual identification for data collection. At the start of every session, we filled three puzzles in sight of the otters to demonstrate that there was food inside to motivate the otters to extract it.

Puzzles were presented *ca* 1 h before a feed to ensure that otters were both active and food motivated. Puzzle trials lasted until all puzzles were 'solved', i.e. the food reward was extracted, or until 20 min had elapsed. Each puzzle session was recorded with two Panasonic video cameras (resolution 1920 × 1080p) positioned around the circumference of the enclosure. An additional camera (Fujifilm Finepix HS20 EXR) was used freehand to facilitate the identification of each otter (see the electronic supplementary material, videos S3–S5). From each puzzle trial video, we scored each otter's: (i) latency to first interact with a puzzle; (ii) time spent interacting with a puzzle until the otter solved it; and (iii) total time from puzzles being dropped into the enclosure until the otter solved a puzzle. We noted if an otter did not solve the puzzle, in which case the 'total time' was the same as the trial duration (1200 s).

## 2.5. Statistical analyses

### 2.5.1. Rock juggling observational data

To address whether otter species, age or sex were associated with differences in rock juggling frequency, we first summed all observations of rock juggling to provide a total count of rock juggling observations

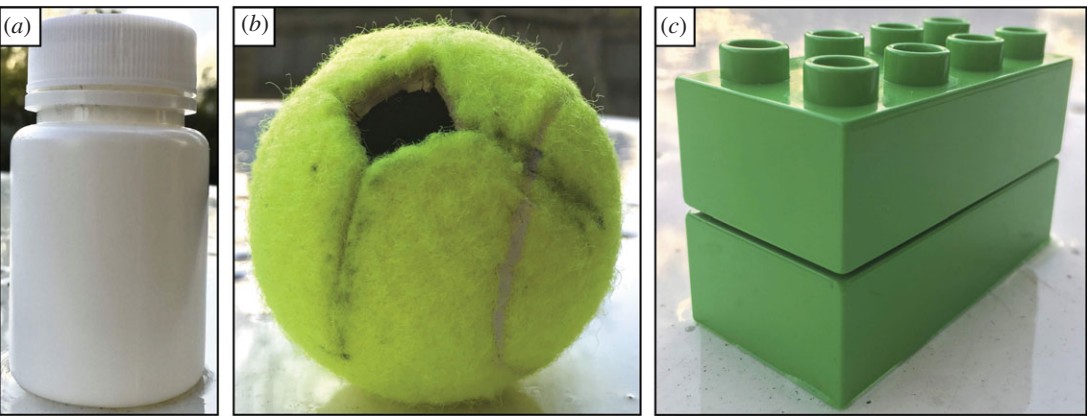

**Figure 1.** The three novel extractive food puzzle types presented to each otter group: (*a*) medicine bottle, (*b*) tennis ball and (*c*) Duplo® bricks. Each task was filled with a food reward.

for each otter. The total number of behaviour scans per otter (regardless of behaviour performed) was also noted, as we could not observe each individual for a total of 252 times (i.e. every 3 min for 12 h, starting at 0 min). Missing observations were owing to unforeseen circumstances, such as the transfer of an individual to another zoo. The otter with the lowest observation count of less than 230 behaviour scans was removed from the sample. An extreme outlier with an unusually high propensity to rock juggle (i.e. a total count of 59 while across all others, average ± s.d. = 12.77 ± 14.95) was also removed from analyses. Asian small-clawed otters at the New Forest Wildlife Park were considerably older (all over 11 years old) than Asian small-clawed otters at both Newquay Zoo and Tamar Otter and Wildlife Park (all less than or equal to 11 years old). To account for this confounding factor in analyses, we categorized otters as either 'young' (less than or equal to 11 years) or 'old' (over 11 years). This age threshold was informed by the 11 year average life expectancy of captive Asian small-clawed otters [45] and the reproductive status of the otters to reflect the senescent individuals (i.e. over 11 years old) in the sample.

To test how species, age and sex were associated with rock juggling frequency, we used a negative binomial generalized linear model as clumping in the count data resulted in an over-dispersed Poisson model. An 'offset' for total observation count was included to inform the model of variance in observation effort. Rock juggling count was the response variable and species, age, age category, sex and site were the fixed effects. Site was included in the model to account for variation in otter behaviour between the zoos. The model also included interactions between age and age category, and age and site. Stepwise backwards elimination of least significant terms was conducted to produce a minimal adequate model (MAM). For each non-significant predictor, the *z*- or *t*- and *p*-values are reported from the model summary before it was removed from the model, while for significant predictors, we also report the slope estimates ± s.e. from the MAM summary. For each significant predictor, a likelihood ratio test using command 'anova' was run to compare the MAM to a model where that fixed effect was removed. These ANOVAs provided the likelihood ratio test/$\chi^2$ and *p*-values reported and indicate whether there is a statistical difference in the amount of deviance explained by the two models including versus excluding the predictors of interest.

To test whether hunger levels (as inferred by time since last feed) were associated with differences in juggling frequency, we summed the number of rock juggling observations when otters were assumed to be hungry versus full, as well as the total number of behaviour scans when otters were hungry/full. A negative binomial generalized linear mixed-effects model was run that offset total observation count, had rock juggling as the response variable, hunger level as the fixed effect and individual ID as a random effect. Model selection was conducted as described above.

### 2.5.2. Den camera trap data

The camera trap generated 556 usable 15 s video clips. To test whether hunger levels were associated with rock juggling frequency in Newquay Zoo otters while they spent time inside, we used known feeding times and the time of the recording to infer whether otters were relatively hungry or full by calculating time since last feed. As individuals could not be identified, Pearson's $\chi^2$ test was run on a

contingency table of rock juggling occurrences versus absences when hungry versus full. Observed values were then compared to expected values to assess the direction of the effect.

To test whether pup presence influenced rock juggling frequency in adults (e.g. because adults might demonstrate rock juggling to their pups), Pearson's $\chi^2$ test was run on a contingency table of rock juggling occurrences versus absences when pups were present versus absent. Observed values were then compared to expected values to assess the direction of the effect.

To control for confounding factors, the influence of time since last feed (i.e. 'hunger') on pup presence in the den was assessed. A Pearson's $\chi^2$ test was run on a contingency table of pup presence versus absence when otters were relatively hungry versus full. Observed values were compared to expected values to assess the direction of the effect.

### 2.5.3. Novel extractive food puzzle data

To address whether rock juggling frequency was associated with extractive foraging performance on the food puzzles, we first removed an outlier from the dataset. This outlier represented an individual that was present in the enclosure but did not interact with any of the tennis ball puzzles for the duration of the trial and thus had a latency of 1200 s, while all other otters had latencies of 90 s or less. For some video trials, individuals could not be identified or moved out of camera shot and so did not have solving success, latencies or interaction times accurately recorded. Such instances generated missing values or 'NAs', which were removed from the dataset before analyses, resulting in 105 usable data points. A subset of 17 puzzle trial videos (i.e. 16.19% of all puzzle trials) were transcribed by two observers blind to the individual otters' rock juggling rates. Inter-observer agreement for latency to first interaction with a food puzzle, duration of puzzle interactions and total time from the start of puzzle presentation until puzzle solve (for puzzle solve successes and fails) was very high (intraclass correlation coefficient (ICC), latency: ICC = 0.94, $n = 17$, $p < 0.001$; interaction time: ICC = 0.98, $n = 17$, $p < 0.001$; total time: ICC = 1, $n = 17$, $p < 0.001$).

As rock juggling count was used as a fixed effect in analyses to predict puzzle-solving performance, an offset could not be used to control for variance in observer effort. Instead, a juggle rate was calculated by dividing the total count of rock juggling by the total number of observation points for each otter. M-L. Allison 2018, personal observation, suggested that juggle rate differed between species. We therefore included a juggle rate–species interaction in the statistical models when this did not cause major model convergence issues, as described below.

To test whether latency to first interaction with each food puzzle type was shorter for younger, and possibly more explorative, otters, and/or for those who rock juggled more frequently, we used a linear mixed-effects model. The response variable was each otter's latency to first interact with each food puzzle, fixed effects were juggle rate, species and their interaction, age, sex, site, puzzle type, puzzle presentation order and their interaction, and random effects were individual ID nested within group ID. As plotting the data suggested that the species showed large differences in their latency to start interacting with the different puzzle types, we explored these patterns further by running the same model again, but on the data for each species separately.

Solving success was noted as a binary category with '1' representing success while '0' represented an otter's failure to solve a puzzle (i.e. extract the food reward) during the trial. To test whether otters that rock juggled at a higher frequency were more successful at solving puzzles, we initially used a generalized linear mixed-effects model with binomial error structure. Solving success was the response variable, fixed effects were juggle rate, species and their interaction, latency to first interaction with the task, age, sex, site, puzzle type, puzzle presentation order and their interaction, and individual ID nested within group ID were random effects. However, this model failed to converge until we simplified it to include only the main predictors of interest: species, juggle rate and puzzle type.

To test whether otters that rock juggled more frequently required less interaction time to solve puzzles, we used a linear mixed-effects model on data including only puzzle solves, and excluding puzzle solve failures to avoid ceiling effects. Interaction time was the response variable, juggle rate, species, the interaction between juggle rate and species, age, sex, site, puzzle type, puzzle presentation order and the interaction between puzzle type and puzzle presentation order were fixed effects, and individual ID nested within group ID were random effects.

Model selection was conducted as described above for the rock juggling observational data. We conducted all statistical analyses in RSTUDIO v. 3.6.0 [46].

The interaction between puzzle type and puzzle presentation order appeared to be a significant predictor for latency to first interaction with puzzles ($\chi^2$ test: $\chi^2_4 = 57.352$, $p < 0.001$), time spent interacting with puzzles ($\chi^2$ test: $\chi^2_4 = 25.244$, $p < 0.001$) and time spent interacting with puzzles before successfully solving them ($\chi^2$ test: $\chi^2_4 = 77.097$, $p < 0.001$), but was not a significant predictor for solving success ($\chi^2$ test: $\chi^2_4 = 8.986$, $p = 0.061$). However, Tukey *post hoc* tests revealed that these significant results were owing entirely to extended latency and interaction times in smooth-coated otters that were presented tennis balls first. None of the other comparisons were significant. We report the associated statistics in the electronic supplementary material (tables S4–S6) and have omitted puzzle order from all models described in the main text.

# 3. Results

## 3.1. Do otter species differ in rock juggling frequency?

Smooth-coated otters appeared to rock juggle less ($n = 6$, median count [lower quartile range (LQR)–upper quartile range (UQR)] = 4 [2–6.75]) than Asian small-clawed otters ($n = 42$, median count [LQR–UQR] = 8 [2–18]), but including 'species' did not significantly improve the model fit to the data (species: $z = -1.800$, $p = 0.072$; negative binomial likelihood ratio test (LRT: $\chi^2_1 = 2.889$, $p = 0.089$) (figure 2).

## 3.2. Does otter age correlate with rock juggling frequency?

Overall, rock juggling frequency appeared to increase with increased age, with otters in the 'young' age category (i.e. less than or equal to 11 years old; $n = 38$, median count [LQR–UQR] = 5 [2–12]) demonstrating less rock juggling than otters in the 'old' age category (i.e. over 11 years old; $n = 10$, median count [LQR–UQR] = 19.5 [9.75–34.75]). Age and age category individually did not significantly correlate with rock juggling frequency (continuous age: $z = 1.174$, $p = 0.240$; age category: $z = 1.159$, $p = 0.247$) and did not improve the model fit (negative binomial LRT: $\chi^2_2 = 1.196$, $p = 0.274$). However, including the interaction between age and age category significantly improved the model fit to the data when compared with the null model (negative binomial LRT: $\chi^2_2 = 24.044$, $p < 0.001$; rock juggling frequency in 'young' otters significantly decreased with increasing age (slope estimate ± s.e. = $-0.27 \pm 0.06$, $z = -4.743$, $p < 0.001$). However, in 'old' otters, rock juggling increased with increasing age but not significantly so (slope estimate ± s.e. = $0.01 \pm 0.03$, $z = 0.476$, $p = 0.634$) (figure 3). Any further influence of age was not found to be associated with site (age–site interaction: negative binomial LRT: $\chi^2_2 = 4.899$, $p = 0.086$). The den camera trap footage suggests that age-related group composition may also influence rock juggling frequency: adult otters at Newquay Zoo rock juggled significantly less than expected when pups were present (12 out of 387 video clips) than when pups were absent (13 out of 169 video clips) ($\chi^2$ test: $\chi^2_1 = 4.755$, $p = 0.029$).

## 3.3. Do otter males and females differ in rock juggling frequency?

Females appeared to rock juggle more frequently than males in both Asian small-clawed otters (females: $n = 23$, median count [LQR–UQR] = 11 [4.5–19.5]; males: $n = 19$, median count [LQR–UQR] = 5 [2–16]) and smooth-coated otters (females: $n = 4$, median count [LQR–UQR] = 4.5 [2–9.75]; males: $n = 2$, median count [LQR–UQR] = 3.5 [2.25–4.75]). However, including sex did not significantly improve the model fit to the data (sex: $z = -1.542$, $p = 0.123$; negative binomial LRT: $\chi^2_1 = 2.416$, $p = 0.120$).

## 3.4. Is hunger associated with rock juggling frequency?

Otters rock juggled significantly more 2 h after feeding, i.e. when 'hungry' ($n = 48$, median count [LQR–UQR] = 4.5 [1–12.25]) than they did less than 2 h after feeding, i.e. when 'satiated' ($n = 48$, median count [LQR–UQR] = 1 [0–4.25]) (hunger: slope estimate ± s.e. = $0.40 \pm 0.18$, $z = 2.208$, $p = 0.027$; $\chi^2$ test: $\chi^2_1 = 4.761$, $p = 0.029$) (figure 4). However, observations from the camera trap installed at Newquay Zoo showed that in the den, adult otters did not rock juggle more than expected when 'hungry' (17 out of 383 video clips) than when 'full' (8 out of 173 video clips) ($\chi^2$ test: $\chi^2_1 < 0.001$, $p = 1$). This result did not appear to be confounded by pup presence, as pups were not present more than expected when 'hungry' (269 out of 383 video clips) than when 'full' (118 out of 173 video clips) ($\chi^2$ test: $\chi^2_1 = 0.160$, $p = 0.703$). However, as individuals could not be ID-ed from the video footage, it is unclear

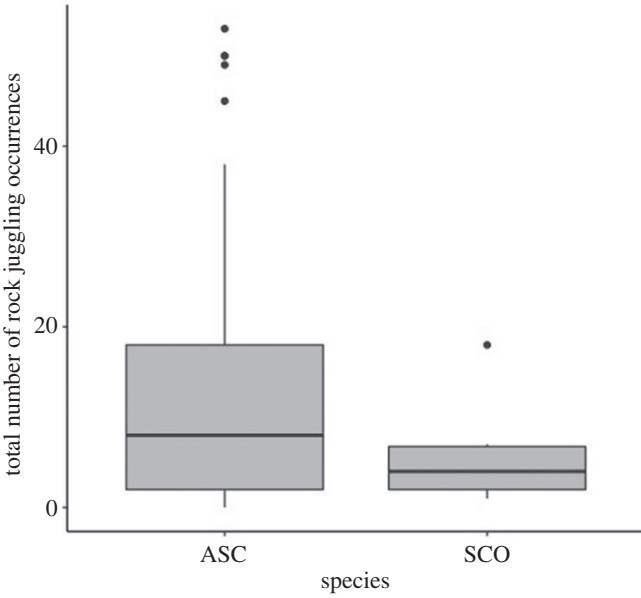

**Figure 2.** Total number of rock juggling occurrences for Asian small-clawed (ASC) and smooth-coated otters (SCO). The dark lines represent medians, the grey box indicates the interquartile range (IQR), whiskers are 1.5 times the IQR and outliers are represented by black circles.

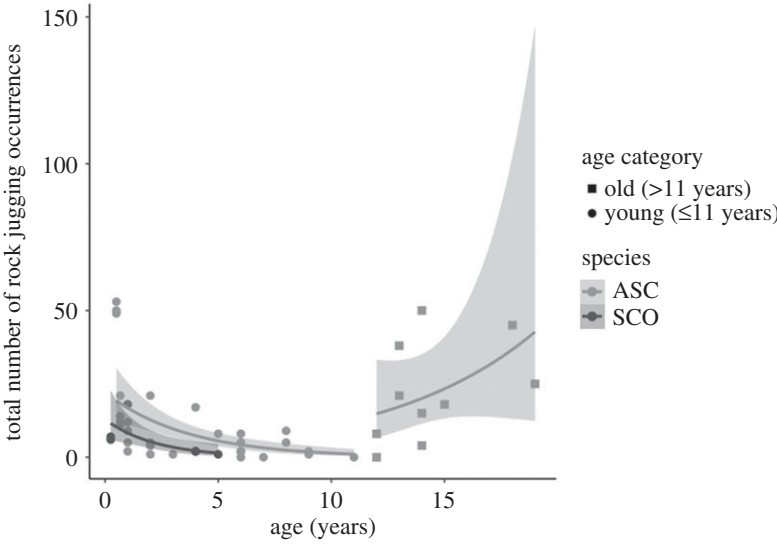

**Figure 3.** Rock juggling occurrences change with age in different directions in 'young' (less than or equal to 11 years old) versus 'old' otters (over 11 years old): while rock juggling decreases with age in young ASC and SCO otters, it may increase with age in old ASC otters. Negative binomial regression lines have been fitted with a shaded 95% confidence interval ribbon for each age category and species.

to what extent these results are driven by a small number of individuals (e.g. the reproductive adults in the group) performing rock juggling in the den.

## 3.5. Does rock juggling frequency correlate with food extraction skill?

Smooth-coated otters took significantly longer to start interacting with puzzles ($n = 6$, mean ± s.e. = 12.71 ± 5.86 s) than Asian small-clawed otters ($n = 39$, mean ± s.e. = 2.27 ± 0.40 s) (species: slope estimate ± s.e. = 9.97 ± 2.95, $t = 3.377$, $p = 0.010$; $\chi^2$ test: $\chi^2_1 = 8.481$, $p = 0.004$). However, latency to smooth-coated otters' first interaction with the food puzzles differed significantly among puzzle types ($\chi^2$ test: $\chi^2_2 = 9.372$, $p = 0.009$), while Asian small-clawed otters showed no difference in latency to interact with

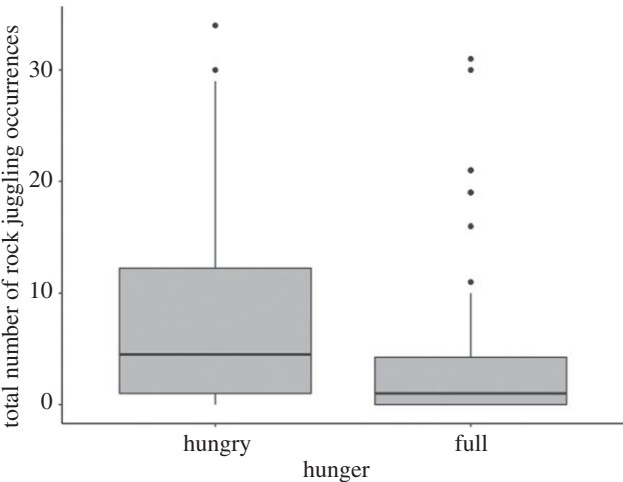

**Figure 4.** Total number of rock juggling occurrences for otters when 'hungry' and 'full'. The dark lines represent medians, the grey boxes indicate the interquartile range (IQR), whiskers are 1.5× times the IQR and outliers are represented by black circles.

the different puzzle types ($\chi^2$ test: $\chi^2_2 = 0.405$, $p = 0.817$). Tukey *post hoc* tests revealed that smooth-coated otters took significantly longer than Asian small-clawed otters to start interacting with tennis balls (SCO: $n = 6$, mean ± s.e. = 33.33 ± 13.39 s; ASC: $n = 25$, mean ± s.e. = 2.12 ± 0.69 s, $p < 0.001$), but there was no significant species difference in latency to start interacting with bottles (SCO: $n = 5$, mean ± s.e. = 1.60 ± 0.68 s; ASC: $n = 36$, mean ± s.e. = 2.47 ± 0.77 s, $p = 1$) or bricks (SCO: $n = 6$, mean ± s.e. = 1.33 ± 0.61 s; ASC: $n = 27$, mean ± s.e. = 2.15 ± 0.49, $p = 1$) (figure 5).

The length of time otters spent interacting with puzzles (regardless of whether the puzzle was solved or not) was significantly correlated with juggle rate ($\chi^2$ test: $\chi^2_1 = 4.253$, $p = 0.039$), age ($\chi^2$: $\chi^2_1 = 5.696$, $p = 0.017$) and puzzle type ($\chi^2$: $\chi^2_2 = 14.969$, $p < 0.001$); more frequent jugglers spent more time interacting with the puzzles ($n = 45$, slope estimate ± s.e. = 635.47 ± 298.03, $t = 2.132$, $p = 0.037$); as otter age increased, puzzle interaction times decreased ($n = 45$, slope estimate ± s.e. = −10.65 ± 4.59, $t = −2.319$, $p = 0.034$); and otters interacted most with bricks ($n = 33$, mean ± s.e. = 251.12 ± 51.83 s), followed by tennis balls ($n = 31$, mean ± s.e. = 116.32 ± 40.92 s) and least with bottles ($n = 41$, mean ± s.e. = 63.02 ± 10.69 s).

Overall, individuals' juggle rate did not predict their puzzle-solving success ($z = −0.294$, $p = 0.768$; $\chi^2$ test: $\chi^2_1 = 0.087$; $p = 0.768$), nor did it predict the length of time they spent interacting with puzzles prior to solving them ($t = −0.098$, $p = 0.923$; $\chi^2$ test: $\chi^2_1 = 0.010$, $p = 0.920$). Smooth-coated otters tended to be less successful at solving puzzles (35.29% ($n = 6$, 6 out of 17) of puzzles) than Asian small-clawed otters (67.05% ($n = 39$, 59 out of 88) of puzzles) but not significantly so (slope estimate ± s.e. = −1.63 ± 0.79; $z = −2.047$, $p = 0.041$; $\chi^2$ test: $\chi^2_1 = 3.305$, $p = 0.069$). Smooth-coated otters also spent significantly longer interacting with the puzzles before successfully solving them ($n = 3$, mean ± s.e. = 196.17 ± 126.96 s) than did Asian small-clawed otters ($n = 34$, mean ± s.e. = 51.78 ± 8.90 s, slope estimate ± s.e. = 144.39 ± 46.90, $t = 3.078$, $p = 0.003$; $\chi^2$ test: $\chi^2_1 = 8.852$, $p = 0.003$).

## 4. Discussion

This study is, to our knowledge, the first to investigate object play behaviour in smooth-coated otters and offers a more detailed analysis of object play behaviour in Asian small-clawed otters. We found that rock juggling frequency decreased with age in mature otters, while it appeared to increase with age in relatively elderly individuals. We found no significant species or sex differences in rock juggling frequency, but otters were rock juggling significantly more before feeds than when they were satiated. Finally, we hypothesized that rock juggling might be related to foraging dexterity and may enhance an individual's ability to extract foods, but our food puzzle findings did not support this hypothesis.

We predicted that Asian small-clawed otters would rock juggle more than smooth-coated otters as they consume crustaceans and molluscs (foods that require extraction), whereas smooth-coated otters are primarily piscivorous [35], but our results did not reflect this. It should be noted that our sample size for smooth-coated otters was limited to only six individuals, making it difficult to confidently ascertain patterns, or lack thereof, in the data. However, we observed that the species differed in *how* they manipulated objects; Asian small-clawed otters performed rapid, intricate movements between

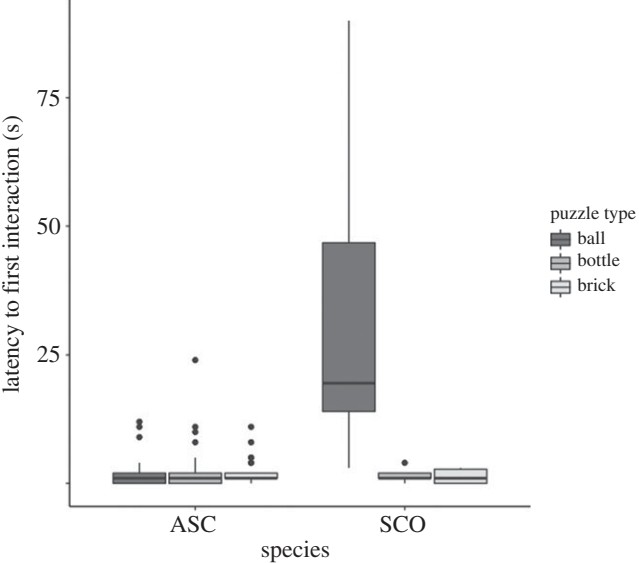

**Figure 5.** Latency, in seconds, until first interaction with puzzle balls, bottles and bricks for Asian small-clawed (ASC) and smooth-coated otters (SCO). The dark lines represent medians, the grey box indicates the interquartile range (IQR), whiskers are 1.5× times the IQR and outliers are represented by black circles.

their forepaws and close to the body, whereas smooth-coated otters batted and threw objects with the forepaws (see the electronic supplementary material, video S6). These dietary and potential object play differences are mirrored in a pair of sympatric African otter species: African clawless otters (*Aonyx capensis*) eat more crustaceans and have been observed to intricately manipulate and juggle objects, while piscivorous spotted-necked otters (*Hydrictis maculicollis*) throw objects [27,47]. As they consume extractive foods, Asian small-clawed and African clawless otters must have high levels of fine motor control [32], an attribute observed in their rock juggling behaviour. However, smooth-coated and spotted-necked otters do not require these intricate movements when foraging and toss items into the air when playing, a behaviour also seen when handling prey [48]. Owing to the large difference in the numbers of smooth-coated and Asian small-clawed otters we had access to, it would be beneficial to extend this study by increasing the sample size of smooth-coated otters, although this is challenging, given their relative rarity in captivity. A detailed investigation of the motor composition and complexity of rock juggling behaviour in different otter species and contexts could provide useful insights into interspecific differences that may relate to ecological traits, as well as intraspecific age and sex-related differences in the manifestation of object play behaviour.

In concurrence with previous studies on the effect of age on object play behaviour [37,49], we found that rock juggling frequency decreased with increasing age in 'young' (i.e. less than or equal to 11 years old) Asian small-clawed otters. During infancy and as juveniles, individuals undergo a crucial period of physical, hormonal and social development [50]. They explore and manipulate objects while in the protection of their parents, allowing them to gather information about their environment which could aid survival in later life and increase fitness [51]. However, studies testing this 'practice hypothesis' have generated conflicting results. While increased social play in juvenile Belding's ground squirrels was suggested to improve motor skills [14], increased play fighting in meerkats (*Suricata suricatta*) does not correlate with winning play fights, nor does winning of play fights or play fighting frequency predict success at gaining dominance [52].

Upon maturity, any developmental benefits of play are no longer garnered [39], and fitness maximizing activities, such as foraging, evading predators and reproducing dominate an animal's available energy and time budgets, placing constraints on their capacity to play [50]. In accordance, we found that adult otters rock juggled less than expected when pups were present (at least in Newquay Zoo's den), but rock juggling frequency increased in old (i.e. over 11 years) Asian small-clawed otters. Martin & Bateson [53] found that domestic cat mothers (*Felis catus*) that had lactation experimentally suppressed performed increased maternal care as kittens spent more time suckling or attempting to suckle. Consequently, experimental mothers played less than control mothers. Accordingly, we hypothesize that the elderly otters of New Forest Wildlife Park had surplus energy and time that could be allotted to object play as they were not

reproducing and had no parental responsibilities [54]. Similarly, Nahallage & Huffman [37] found that senior Japanese macaques (*Macaca fuscata*) also engaged in increased levels of object play. They suggested that, as macaques experience cognitive decline with increased age [55], the adaptive function of the behaviour may be to psychologically relax and maintain neural pathways to delay cognitive decline, just as reading and puzzle-solving prevent the development of mild cognitive impairment in humans [56]. Otters have enlarged areas in their somatic sensory cerebral cortex; in *Aonyx*, this expansion is correlated with the forelimb [57]. It is possible that the repetitive fine motor movements of rock juggling strengthen neural connections [58] within the sensorimotor regions of the brain and delay cognitive decline. In future, a longitudinal study on object play in otters could assess how the behaviour and its related benefits change over an individual's lifetime. As the presence of pups may impact rock juggling behaviour in adults, it would also be worthwhile to investigate the effect of experimentally manipulated group composition on object play behaviour.

Generally, play is thought to decline in individuals when experiencing energetic stress [59–61]. Empirical support for this hypothesis was provided by Sharpe *et al.* [62], who conducted a short-term provisioning experiment on meerkat pups and demonstrated that, when provisioned pups become satiated and were relinquished of the need to beg, rates of play increased. Similarly, Japanese macaques that were food-provisioned multiple times a day demonstrated increased frequency, length and prevalence of stone handling behaviour during provisioned periods when compared with unprovisioned periods [63]. By contrast, we found that rock juggling increased with time since last feed, i.e. when 'hungry', concurring with Pellis [29] who suggested that the captive Asian small-clawed otters he studied were demonstrating 'misdirected foraging'. However, other species may not exhibit any association between hunger levels and object play; in juvenile mink (*Neovison vison*), an apparent diurnal patterning of object play was considered to be a side effect of variations in overall general activity and so was not found to be stimulated by hunger [64]. As such, while the misdirected foraging hypothesis may explain diurnal patterning of object play in otters, it does not explain all object play behaviour demonstrated by other carnivore species.

Appetitive, anticipatory behaviours are common in captive carnivores [65] and captive otters are commonly reported to perform inappropriate anticipatory or begging behaviours before feeding [66]. When investigating the proximate cause of begging behaviour in Asian small-clawed otters, Gothard [67] proposed that rock juggling behaviour may be another form of oral stereotypy. Stereotypies are defined as 'repetitive, unvarying and apparently functionless behaviour patterns' [65, p. 103]. However, Pellis [68] found that the action patterns seen in rock juggling by Asian small-clawed otters featured less manual and more oral manipulations with an increase in object play frequency. Given this variety in behaviour patterns, rock juggling does not appear to be strictly stereotypical and may be more akin to play. Nahallage & Huffman [69] found that stone handling frequency in Japanese macaques declined under stressful environmental conditions (e.g. adverse weather and following veterinary procedures), and increased under relaxed environmental and social conditions. If so, rock juggling could be beneficial to welfare and/or indicative of positive psychological well-being. Further studies combining Pellis' and Nahallage & Huffman's methodologies by assessing the composition and frequency of rock juggling behaviour under varying levels of environmental stress, as well as observing wild populations to ascertain whether this behaviour is confined to captivity, would help elucidate whether this behaviour is object play or stereotypy. Conducting feed enrichment experiments, such as live-feeding of invertebrates, a randomized feeding schedule and removal of keepers associated with feeding [34,67,70], could help determine the welfare impacts and motivations of rock juggling and inform revisions to husbandry methods to prevent the behaviour if deemed necessary.

Juggle rate did not predict an otter's ability to solve food extraction puzzles, suggesting that rock juggling does not enhance food extraction ability. However, these findings need to be interpreted with caution for several reasons. Firstly, it is difficult to disentangle an individual's cognitive ability and their physical capability [71]. As such, an otter may have had the fine motor control needed to extract the food but lacked the knowledge and understanding of how to solve the puzzle. Many of the senior otters were arthritic, which may have disguised potential dexterity-related benefits of rock juggling behaviour and may also explain why older otters spent less time interacting with the food puzzles. Secondly, as the food puzzles were designed to test food extraction ability, they were naturally biased towards the extractive foraging behaviour of Asian small-clawed otters and against the non-extractive foraging behaviour of smooth-coated otters. Thirdly, the use of man-made novel puzzles may have resulted in neophobic behaviour [72]. Smooth-coated otters took significantly longer to initiate interactions with the tennis ball puzzles, which may have been owing to their potentially aversive neon colour. Fourthly, the minced meat reward may not have been valuable enough to warrant engaging with these unfamiliar, and potentially risky, objects [73,74].

Smooth-coated otters demonstrated longer interaction times before solving puzzles than Asian small-clawed otters, which may be because they were motivated by play itself rather than food. Their boisterous handling of the puzzles sometimes resulted in a puzzle breaking open, but instead of eating the food reward, there were occasions when they continued to play with the empty puzzle. This explanation is supported by the finding that frequent rock jugglers also spent more time interacting with the food puzzles. Owing to these confounding factors, it would be premature to reject the proposed 'extractive foraging' hypothesis to explain rock juggling behaviour in otters. It would be beneficial to replicate this study with naturally occurring extractive foods such as mussels or clams, while ensuring that prior experience with these foods is controlled for and that otters acknowledge these foods as edible so that they are motivated to open them. Presenting these extractive foods to otters in solitude would allow extraction of food without interruption by conspecifics and may provide more accurate data on food extraction dexterity.

## 5. Conclusion

Rock juggling was performed with greater frequency prior to feeding but did not appear to be related to food extraction ability. The behaviour could be explained by the 'misdirected foraging' hypothesis and may not have an ultimate function, which could indicate a stereotypic behaviour. Detailed observations of wild otters in their natural habitat are required to ascertain whether rock juggling is confined to captive conditions. However, the different hypotheses for rock juggling in otters may not be mutually exclusive. It is possible that rock juggling could also be explained by motor development in young otters and/or prevention of cognitive decline in senior otters—longitudinal studies of rock juggling frequency and function across individuals' lifetimes in both captivity and the wild would help test these hypotheses. In conclusion, the function of object play behaviour may depend on context, sex and species, and may change across an individual's lifetime. It is important to perform more detailed behavioural studies that consider the form and composition of play as well as the frequency, to navigate the fine line between object play and stereotypy to protect and enhance the welfare of captive animals.

Ethics. Permission to conduct this research was granted by the College of Life and Environmental Sciences Ethics Committee of the University of Exeter and by the New Forest Wildlife Park, Newquay Zoo and Tamar Otter and Wildlife Centre. Experiments were carried out in accordance with the ASAB/ABS Guidelines for the Use of Animals in Research [75].

Data accessibility. Data and code for rock juggling frequency, hunger and food puzzles can be accessed through the Dryad Digital Repository: https://doi.org/10.5061/dryad.rn8pk0p64 [76]. Tables of *post hoc* test results for puzzle order have been uploaded as part of the electronic supplementary material.

Authors' contributions. M-L.A. and R.R. designed the experiment, collected observational and experimental data, and collated all data. E.M. scored behaviours from the den camera trap footage. M-L.A. analysed the data and wrote the manuscript with support from R.R. and E.M. N.J.B. conceived the original idea, supervised the project and critically edited the manuscript.

Competing interests. The authors declare that they have no competing interests.

Funding. N.J.B. is funded by a Royal Society Dorothy Hodgkin Research Fellowship.

Acknowledgements. We give thanks to the otter keepers at the New Forest Wildlife Park, Newquay Zoo and Tamar Otter and Wildlife Centre for kindly allowing us to conduct our study. We thank Tamar Lennard for her participation in data collection and Alex Saliveros for connecting us with study sites and his help in IDing otters. We gratefully acknowledge Erik Postma and Kelly Moyes for providing excellent statistical advice and support. We are also grateful to two anonymous reviewers and the editors for their helpful and constructive comments on the manuscript.

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
