## [Reviewer comments · Royal Society Open Science]

Review History

RSOS-200141.R0 (Original submission)

Review form: Reviewer 1

Is the manuscript scientifically sound in its present form?

Yes

Are the interpretations and conclusions justified by the results?

Yes

Is the language acceptable?

Yes

Do you have any ethical concerns with this paper?

No

Have you any concerns about statistical analyses in this paper?

No

Recommendation?

Accept with minor revision (please list in comments)

Comments to the Author(s)

The paper presents novel data on the object play of two species of otters. The species differ in foraging habits and are used to test several hypotheses about the occurrence of playing with rocks. The use of multiple groups from multiple locations enhances the strength of the conclusions drawn. For example, by comparing 'hungry' and 'sated' otters, the data strongly support the 'object play as misdirected foraging hypothesis'. While this is important because it offers a mechanistic means of explaining the behavior, it should be noted that there are species differences more broadly, suggesting that this may not explain all cases of object play in carnivores. A study on captive mink showed that the frequency of object play is not correlated with hunger (Ahloy Dallaire, J., & Mason, G. J. (2016). Play in juvenile mink: litter effects, stability over time, and motivational heterogeneity. *Developmental Psychobiology*, 58, 945–957). That is, while misdirected foraging may account for the play of some species (as reviewed by Hall (1998), already cited in the paper), it may not explain all cases of species using foraging-typical behavior patterns in their play (see Pellis, S. M., Pellis, V. C., Pelletier, A., & Leca, J.-B. (2019). Is play a behavior system, and, if so, what kind? *Behavioural Processes*, 160, 1-9). As the authors note, Lutrinae are an excellent model taxon for studying object play. Hopefully, the present paper will stimulate more work on otters. The study is well designed and the analyses appropriate. I have no substantive concerns, although along with some minor points of clarification, I do note some limitations in the present data that should be taken into account when discussing the relevance of the findings (see below).

Page 10, line 44: Define "NA"

Page 15, line 1: Regarding species differences – the conclusion that there are no species differences is not convincing. Irrespective of what the model shows, in the Results (page 12), it is noted that SCOs tended to juggle less than ASCs. Moreover, given the sample size discrepancy (see pages 5-6, under test subjects: while the sample of small-clawed otters was large [eight groups, ranging from one individual to 12, for a total of 42 individuals], that for the smooth-coated otters was less so, with only two groups sampled - one having 2 and the other 4 individuals), it cannot be concluded that this lack of a species difference is real, as one aberrant SCO group could bias the outcome. I would prefer to see a bar graph with error bars that show the data for the two species, then in the Discussion, be more tentative about the conclusion, and so leave it to readers to decide whether the data are sufficient to draw a firm conclusion. The authors attenuate their conclusion of a lack of a species difference later in the paragraph, but by then, the overstatement has taken hold. Reverse the order – qualifier first, then a tentative conclusion on what can be drawn from the data available. Also, the qualitative features of rock juggling may differ between the species, an issue not dealt with in the present paper. Even if the frequency of rock juggling may be similar between the two species, the complexity of the movements involved could be different. Given the preceding discussion in the paragraph, the ASCs may well have more complex patterns of rock juggling than SCOs. Do the authors have any observations on this matter that would be useful to describe here? If not, this should be noted as a matter for future study.

Page 17, lines 1-12: Related to the content of the actions used during rock juggling: In this paragraph, on the possibility that this represents a stereotypy, it should be noted that, in one study of ASCs, a sequence of unfolding in the action patterns used during object play was noted. As the intensity of the activity increased, the behavior patterns used also changed – starting with probing actions, followed by gathering actions and ending with fragmentation actions (Pellis, S. M. (1983). The frequency and pattern of play behaviour. *Mammalia*, 47, 272-274). That is, the behavior used went from the early to later stages of the forging cycle. Again, placing 'rock juggling' within the broader context of object play and its changes in content over time would provide clues as to whether the behavior was indeed misdirected foraging behavior or a stereotypy. The authors should note these different possibilities and hopefully conduct the appropriate analyses for future publications.

Pages 17-18, last and first paragraph: It should be noted that the food puzzles are biased for the foraging patterns in the small-clawed otters, not the smooth-coated otters (see page 7, lines 19-49).

Review form: Reviewer 2

Is the manuscript scientifically sound in its present form?

Yes

Are the interpretations and conclusions justified by the results?

Yes

Is the language acceptable?

Yes

Do you have any ethical concerns with this paper?

No

Have you any concerns about statistical analyses in this paper?

No

Recommendation?

Accept with minor revision (please list in comments)

Comments to the Author(s)

This is an interesting paper contrasting the rock juggling play. The limitations of the study make the conclusions more suggestive than definitive, but will be useful in future research as well as theorizing in the animal play field.

Some comments on the study:

- 1) Issue of interobserver reliability and blind testing need to be addressed explicitly,
- 2) How low resolution were the videos and are their samples on line to view/
- 3) What does presenting the puzzle boxes in random order mean? How done?
- 4) Was puzzle box order entered into the analysis? It should be if not done as order effects can be great in these kind of experiments.
- 5) In Fig. 5 only the tennis ball led to a great difference in latency, so the statement on the species differences should be tempered to only refer to that, not an overall puzzle box difference.
- 6) In terms of discussion the many negative functional claims of Sharpe may merit some attention along with some of the cat object play research such as Bateson's group.
- 7) I am surprised that there was no mention of the many studies on rock knocking in Japanese monkeys carried out by Huffman and colleagues. I think they have relevant data that could be usefully discussed.
- 8) Related to the above, the paper ends abruptly with no general discussion that puts the study into the larger context of object play research. Some important papers are presented in the introduction, but their needs to be some integration with this literature given the findings reached in this study.

Decision letter (RSOS-200141.R0)

04-Mar-2020

Dear Miss Allison

On behalf of the Editors, I am pleased to inform you that your Manuscript RSOS-200141 entitled "The drivers and functions of rock juggling in otters" has been accepted for publication in Royal Society Open Science subject to minor revision in accordance with the referee suggestions. Please find the referees' comments at the end of this email.

The reviewers and handling editors have recommended publication, but also suggest some minor revisions to your manuscript. Therefore, I invite you to respond to the comments and revise your manuscript.

- Ethics statement

- Data accessibility

If you wish to submit your supporting data or code to Dryad (<http://datadryad.org/>), or modify your current submission to dryad, please use the following link:
<http://datadryad.org/submit?journalID=RSOS&manu=RSOS-200141>

- Competing interests

- Authors' contributions

AB carried out the molecular lab work, participated in data analysis, carried out sequence alignments, participated in the design of the study and drafted the manuscript; CD carried out the statistical analyses; EF collected field data; GH conceived of the study, designed the study,

coordinated the study and helped draft the manuscript. All authors gave final approval for publication.

- Acknowledgements

- Funding statement

Because the schedule for publication is very tight, it is a condition of publication that you submit the revised version of your manuscript before 13-Mar-2020. Please note that the revision deadline will expire at 00.00am on this date. If you do not think you will be able to meet this date please let me know immediately.

If your manuscript is newly submitted and subsequently accepted for publication, you will be asked to pay the article processing charge, unless you request a waiver and this is approved by Royal Society Publishing. You can find out more about the charges at <https://royalsocietypublishing.org/rsos/charges>. Should you have any queries, please contact openscience@royalsociety.org.

on behalf of Dr Alexander Ophir (Associate Editor) and Pete Smith (Subject Editor)
openscience@royalsociety.org

Associate Editor Comments to Author (Dr Alexander Ophir):
Comments to the Author:
Dear Dr Allison,

I have received reviews from two leading experts in the field, who were both very positive about your paper. I concur with their positive assessment of your manuscript and believe that this will be a very nice addition to the literature on play behavior. Both reviewers provided a list of helpful and fairly minor points, that your manuscript will benefit from if you can integrate or address. In particular, please place careful consideration on the strength of the conclusions you raised, as noted by the reviewers; I believe the appeal of your study will be strengthened taking a more cautious approach to the interpretations of your results. I would like to congratulate you and your co-authors on this paper.

Alex Ophir
Associate Editor, RSOS

Reviewer comments to Author:
Reviewer: 1

Comments to the Author(s)
The paper presents novel data on the object play of two species of otters. The species differ in foraging habits and are used to test several hypotheses about the occurrence of playing with

rocks. The use of multiple groups from multiple locations enhances the strength of the conclusions drawn. For example, by comparing 'hungry' and 'sated' otters, the data strongly support the 'object play as misdirected foraging hypothesis'. While this is important because it offers a mechanistic means of explaining the behavior, it should be noted that there are species differences more broadly, suggesting that this may not explain all cases of object play in carnivores. A study on captive mink showed that the frequency of object play is not correlated with hunger (Ahloy Dallaire, J., & Mason, G. J. (2016). Play in juvenile mink: litter effects, stability over time, and motivational heterogeneity. *Developmental Psychobiology*, 58, 945-957). That is, while misdirected foraging may account for the play of some species (as reviewed by Hall (1998), already cited in the paper), it may not explain all cases of species using foraging-typical behavior patterns in their play (see Pellis, S. M., Pellis, V. C., Pelletier, A., & Leca, J.-B. (2019). Is play a behavior system, and, if so, what kind? *Behavioural Processes*, 160, 1-9). As the authors note, *Lutrinae* are an excellent model taxon for studying object play. Hopefully, the present paper will stimulate more work on otters. The study is well designed and the analyses appropriate. I have no substantive concerns, although along with some minor points of clarification, I do note some limitations in the present data that should be taken into account when discussing the relevance of the findings (see below).

Page 10, line 44: Define "NA"

Page 15, line 1: Regarding species differences – the conclusion that there are no species differences is not convincing. Irrespective of what the model shows, in the Results (page 12), it is noted that SCOs tended to juggle less than ASCs. Moreover, given the sample size discrepancy (see pages 5-6, under test subjects: while the sample of small-clawed otters was large [eight groups, ranging from one individual to 12, for a total of 42 individuals], that for the smooth-coated otters was less so, with only two groups sampled - one having 2 and the other 4 individuals), it cannot be concluded that this lack of a species difference is real, as one aberrant SCO group could bias the outcome. I would prefer to see a bar graph with error bars that show the data for the two species, then in the Discussion, be more tentative about the conclusion, and so leave it to readers to decide whether the data are sufficient to draw a firm conclusion. The authors attenuate their conclusion of a lack of a species difference later in the paragraph, but by then, the overstatement has taken hold. Reverse the order – qualifier first, then a tentative conclusion on what can be drawn from the data available. Also, the qualitative features of rock juggling may differ between the species, an issue not dealt with in the present paper. Even if the frequency of rock juggling may be similar between the two species, the complexity of the movements involved could be different. Given the preceding discussion in the paragraph, the ASCs may well have more complex patterns of rock juggling than SCOs. Do the authors have any observations on this matter that would be useful to describe here? If not, this should be noted as a matter for future study.

Page 17, lines 1-12: Related to the content of the actions used during rock juggling: In this paragraph, on the possibility that this represents a stereotypy, it should be noted that, in one study of ASCs, a sequence of unfolding in the action patterns used during object play was noted. As the intensity of the activity increased, the behavior patterns used also changed – starting with probing actions, followed by gathering actions and ending with fragmentation actions (Pellis, S. M. (1983). The frequency and pattern of play behaviour. *Mammalia*, 47, 272-274). That is, the behavior used went from the early to later stages of the forging cycle. Again, placing 'rock juggling' within the broader context of object play and its changes in content over time would provide clues as to whether the behavior was indeed misdirected foraging behavior or a stereotypy. The authors should note these different possibilities and hopefully conduct the appropriate analyses for future publications.

Pages 17-18, last and first paragraph: It should be noted that the food puzzles are biased for the foraging patterns in the small-clawed otters, not the smooth-coated otters (see page 7, lines 19-49).

Reviewer: 2

Comments to the Author(s)

This is an interesting paper contrasting the rock juggling play. The limitations of the study make the conclusions more suggestive than definitive, but will be useful in future research as well as theorizing in the animal play field.

Some comments on the study:

- 1) Issue of interobserver reliability and blind testing need to be addressed explicitly,
- 2) How low resolution were the videos and are their samples on line to view/
- 3) What does presenting the puzzle boxes in random order mean? How done?
- 4) Was puzzle box order entered into the analysis? It should be if not done as order effects can be great in these kind of experiments.
- 5) In Fig. 5 only the tennis ball led to a great difference in latency, so the statement on the species differences should be tempered to only refer to that, not an overall puzzle box difference.
- 6) In terms of discussion the many negative functional claims of Sharpe may merit some attention along with some of the cat object play research such as Bateson's group.
- 7) I am surprised that there was no mention of the many studies on rock knocking in Japanese monkeys carried out by Huffman and colleagues. I think they have relevant data that could be usefully discussed.
- 8) Related to the above, the paper ends abruptly with no general discussion that puts the study into the larger context of object play research. Some important papers are presented in the introduction, but their needs to be some integration with this literature given the findings reached in this study.

Author's Response to Decision Letter for (RSOS-200141.R0)

See Appendix A.

Decision letter (RSOS-200141.R1)

08-Apr-2020

Dear Miss Allison,

It is a pleasure to accept your manuscript entitled "The drivers and functions of rock juggling in otters" in its current form for publication in Royal Society Open Science. The comments of the reviewer(s) who reviewed your manuscript are included at the foot of this letter.

on behalf of Dr Alexander Ophir (Associate Editor) and Pete Smith (Subject Editor)
openscience@royalsociety.org

Associate Editor Comments to Author (Dr Alexander Ophir):

Comments to the Author:

Dear Dr. Allison,

Thank you for your revised manuscript and for being so responsive to the reviewers and their suggestions. I am happy to recommend that your manuscript be accepted. Congratulations, and I hope you and all involved are doing well during these unprecedented and challenging times.

Alex Ophir
Associate Editor, RSOS

Appendix A

Dear Dr Allison,

I have received reviews from two leading experts in the field, who were both very positive about your paper. I concur with their positive assessment of your manuscript and believe that this will be a very nice addition to the literature on play behavior. Both reviewers provided a list of helpful and fairly minor points, that your manuscript will benefit from if you can integrate or address. In particular, please place careful consideration on the strength of the conclusions you raised, as noted by the reviewers; I believe the appeal of your study will be strengthened taking a more cautious approach to the interpretations of your results. I would like to congratulate you and your co-authors on this paper.

Alex Ophir
Associate Editor, RSOS

Thank you so much for such a positive response to our work. We are grateful for all the constructive comments that have allowed us to refine our manuscript. We agree that we should be more cautious with our conclusions and have rephrased our discussion so we do not overstate our findings.

Reviewer comments to Author:

Reviewer: 1

Comments to the Author(s)

The paper presents novel data on the object play of two species of otters. The species differ in foraging habits and are used to test several hypotheses about the occurrence of playing with rocks. The use of multiple groups from multiple locations enhances the strength of the conclusions drawn. For example, by comparing ‘hungry’ and ‘sated’ otters, the data strongly support the ‘object play as misdirected foraging hypothesis’. While this is important because it offers a mechanistic means of explaining the behavior, it should be noted that there are species differences more broadly, suggesting that this may not explain all cases of object play in carnivores. A study on captive mink showed that the frequency of object play is not correlated with hunger (Ahloy Dallaire, J., & Mason, G. J. (2016). Play in juvenile mink: litter effects, stability over time, and motivational heterogeneity. *Developmental Psychobiology*, 58, 945–957). That is, while misdirected foraging may account for the play of some species (as reviewed by Hall (1998), already cited in the paper), it may not explain all cases of species using foraging-typical behavior patterns in their play (see Pellis, S. M., Pellis, V. C., Pelletier, A., & Leca, J.-B. (2019). Is play a behavior system, and, if so, what kind? *Behavioural Processes*, 160, 1-9). As the authors note, Lutrinae are an excellent model taxon for studying object play. Hopefully, the present paper will stimulate more work on otters. The study is well designed and the analyses appropriate. I have no substantive concerns, although along with some minor points of clarification, I do note some limitations in the present data that should be taken into account when discussing the relevance of the findings (see below).

Thank you for your positive response regarding our work. We too hope that this study will encourage much needed further research into otters.

Thank you for your incredibly detailed and constructive comments. We have revised our phrasing regarding object play as misdirected foraging to ensure that it does not suggest a blanket statement that applies to all carnivores. We have used the reference you kindly provided and have clarified that the misdirected foraging hypothesis cannot account for all cases of foraging-typical behaviour patterns in play; please see lines 690-695.

Page 10, line 44: Define “NA”

We have now clarified which observations were classed as “NA” and why; please see lines 301-304.

Page 15, line 1: Regarding species differences – the conclusion that there are no species differences is not convincing. Irrespective of what the model shows, in the Results (page 12), it is noted that SCOs tended to juggle less than ASCs. Moreover, given the sample size discrepancy (see pages 5-6, under test subjects: while the sample of small-clawed otters was large [eight groups, ranging from one individual to 12, for a total of 42 individuals], that for the smooth-coated otters was less so, with only two groups sampled - one having 2 and the other 4 individuals), it cannot be concluded that this lack of a species difference is real, as one aberrant SCO group could bias the outcome. I would prefer to see a bar graph with error bars that show the data for the two species, then in the Discussion, be more tentative about the conclusion, and so leave it to readers to decide whether the data are sufficient to draw a firm conclusion. The authors attenuate their conclusion of a lack of a species difference later in the paragraph, but by then, the overstatement has taken hold. Reverse the order – qualifier first, then a tentative conclusion on what can be drawn from the data available. Also, the qualitative features of rock juggling may differ between the species, an issue not dealt with in the present paper. Even if the frequency of rock juggling may be similar between the two species, the complexity of the movements involved could be different. Given the preceding discussion in the paragraph, the ASCs may well have more complex patterns of rock juggling than SCOs. Do the authors have any observations on this matter that would be useful to describe here? If not, this should be noted as a matter for future study.

We have now added a box plot to further illustrate any differences between SCOs and ASCs in rock juggling frequency. We chose to use a box plot as the data was not normally distributed. We have also rearranged the species discussion paragraph to ensure that we highlight the sample size discrepancy before drawing conclusions; please see lines 574-576. We have supplied sample videos in the supplementary electronic material depicting the difference in rock juggling behaviour between ASCs and SCOs. However, we do not have any formal observations to add so have included this in suggestions for future study; please see lines 603-606.

Page 17, lines 1-12: Related to the content of the actions used during rock juggling: In this paragraph, on the possibility that this represents a stereotypy, it should be noted that, in one study of ASCs, a sequence of unfolding in the action patterns used during object play was noted. As the intensity of the activity increased, the behavior patterns used also changed – starting with probing actions, followed by gathering actions and ending with fragmentation actions (Pellis, S. M. (1983). The frequency and pattern of play behaviour. *Mammalia*, 47, 272-274). That is, the behavior used went from the early to later stages of the forging cycle. Again, placing ‘rock juggling’ within the broader context of object play and its changes in content over time would provide clues as to whether the behavior was indeed misdirected foraging behavior or a stereotypy. The authors should note these different possibilities and hopefully conduct the appropriate analyses for future publications. ***Thank you for supplying us with this paper so that we could include it in our study. It has greatly enriched the content of our Discussion; please see lines 738-741.***

Pages 17-18, last and first paragraph: It should be noted that the food puzzles are biased for the foraging patterns in the small-clawed otters, not the smooth-coated otters (see page 7, lines 19-49).

We have added a statement to highlight this bias in food puzzles for ASC foraging behaviour; please see lines 813-815.

Reviewer: 2

Comments to the Author(s)

This is an interesting paper contrasting the rock juggling play. The limitations of the study make the conclusions more suggestive than definitive, but will be useful in future research as well as theorizing in the animal play field.

Thank you for your interest in our work and for the kind acknowledgement of the contribution our study may provide to future animal play research.

Some comments on the study:

1) Issue of interobserver reliability and blind testing need to be addressed explicitly,

We have now performed inter-observer correlation coefficient tests and report the results within the statistical analyses section; please see lines 304-310.

2) How low resolution were the videos and are their samples on line to view/

We have now stated the resolution of the videos; please see lines 183-184 and 225.

3) What does presenting the puzzle boxes in random order mean? How done?

We have provided further detail to clarify how puzzles were presented; please see lines 211-213.

4) Was puzzle box order entered into the analysis? It should be if not done as order effects can be great in these kind of experiments.

We have performed analyses including puzzle presentation order, which initially appeared to indicate that presentation order significantly influenced the results. However, following post hoc tests, we found that one group of SCOs was skewing data considerably due to their cautious response to the tennis balls. We have reported these findings in text (please see lines 369-377) and have included tables of the relevant results from the post-hoc tests in the electronic supplementary materials.

5) In Fig. 5 only the tennis ball led to a great difference in latency, so the statement on the species differences should be tempered to only refer to that, not an overall puzzle box difference.

We have reported results from a post hoc test which clarifies that the species difference only lies within the tennis ball puzzle toys; please see lines 450-463.

6) In terms of discussion the many negative functional claims of Sharpe may merit some attention along with some of the cat object play research such as Bateson's group.

We have now referred to Sharpe's study on meerkats which found no evidence that increased play-fighting was correlated with higher success at winning play-fights or fights for higher dominance ranks, to demonstrate the conflicting evidence regarding the “practice hypothesis”; please see lines 615-617. We have also included a reference to Martin and Bateson's study on cats assessing the impacts of lactation suppression on play behaviour in kittens and mothers to inform our hypothesis on decreased object play in adult otters but increased object play in senior adults that did not have juveniles in their group; please see lines 624-627.

7) I am surprised that there was no mention of the many studies on rock knocking in Japanese monkeys carried out by Huffman and colleagues. I think they have relevant data that could be usefully discussed.

We had initially referenced a Nahallage and Huffman paper (please see [37] on reference list and line 111 for first use) as it heavily informed our hypothesis and predictions concerning age differences in rock juggling behaviour of otters (especially the potential proximate function of rock juggling in senior otters). We recognise that Huffman's work on stone handling in Japanese macaques closely aligns with our study on rock juggling in otters. As such, we have referenced additional studies performed by him and his colleagues that we felt were appropriate and informative to our study; please see lines 685-688 and 741-744.

8) Related to the above, the paper ends abruptly with no general discussion that puts the study into the larger context of

object play research. Some important papers are presented in the introduction, but their needs to be some integration with this literature given the findings reached in this study.

We have added closing statements to place our findings in a broader context in line with existing literature concerning object play behaviour; please see lines 851-855.